# Assessing Clinical Outcomes of Vancomycin Treatment in Adult Patients with Vancomycin-Susceptible *Enterococcus faecium* Bacteremia

**DOI:** 10.3390/antibiotics12111577

**Published:** 2023-10-29

**Authors:** Hiroshi Sasano, Kazuhiko Hanada

**Affiliations:** 1Department of Pharmacometrics and Pharmacokinetics, Meiji Pharmaceutical University, 2-522-1 Noshio, Kiyose, Tokyo 204-8588, Japan; hsasano@juntendo.ac.jp; 2Department of Pharmacy, Juntendo Tokyo Koto Geriatric Medical Center, Tokyo 136-0075, Japan

**Keywords:** vancomycin, *Enterococcus faecium*, bacteremia

## Abstract

Purpose: Enterococcal bacteremia is associated with high mortality and long-term hospitalization. Here, we aimed to investigate the clinical outcomes and evaluate the risk factors for mortality in adult patients treated with vancomycin (VCM) for vancomycin-susceptible *Enterococcus faecium* (*E. faecium*) bacteremia. Methods: This is a retrospective, record-based study. The data were collected from inpatients at a single university hospital between January 2009 and December 2020. The area under the curve (AUC) of VCM was calculated using the Bayesian approach. The primary outcome was a 30-day in-hospital mortality. Results: A univariate analysis showed significant differences in the concomitant use of vasopressors, history of the use of no clinically relevant activity antimicrobial agents against *E. faecium*, VCM plasma trough concentration, and renal dysfunction during VCM administration between the 30-day in-hospital mortality and survival groups. However, the groups’ AUC/minimum inhibitory concentration (MIC) were not significantly different. A multivariate analysis suggested that concomitant vasopressors may be an independent risk factor for 30-day in-hospital mortality (odds ratio, 7.81; 95% confidence interval, 1.16–52.9; *p* = 0.035). The VCM plasma trough concentrations and the AUC/MIC in the mortality group were higher than those in the surviving group. No association between the AUC/MIC and the treatment effect in *E. faecium* bacteremia was assumed, because the known, target AUC/MIC were sufficiently achieved in the mortality group. Conclusions: There may be no association between the AUC/MIC and the treatment effect in *E. faecium* bacteremia. When an immunocompromised host develops *E. faecium* bacteremia with septic shock, especially when a vasopressor is used in a patient with unstable hemodynamics, it may be difficult to treat it, despite efforts to ensure the appropriate AUC/MIC and therapeutic vancomycin concentration levels.

## 1. Introduction

The *Enterococcus* species are the Gram-positive facultative anaerobic cocci that constitute the normal bacterial microbiota in human and animal intestines [1]. However, enterococcal bacteremia is associated with high mortality and long-term hospitalization [1,2,3]. Most cases of invasive enterococcal bacteremia are caused by the *Enterococcus faecalis*, followed by the *Enterococcus faecium* (*E. faecium*) [4]. The *E. faecium* is commonly found in the microbiota of the human gut and mainly causes urinary tract infections, wound infections, endocarditis, and bacteremia [5].

Vancomycin (VCM) is used for the treatment of vancomycin-susceptible *E. faecium* infection. To achieve clinical efficacy and maintain minimum toxicity, the area under the serum’s concentration–time curve (AUC)/minimum inhibitory concentration (MIC) ratio of VCM is recommended to be 400–600 [6]. This target ratio is a therapeutic goal for a suspected or definitive diagnosis of MRSA. A previous study suggested that an AUC/MIC ≥ 389 was an independent factor in reducing mortality from enterococcal bacteremia [7]. However, the study evaluated all *Enterococcus* spp. and was not limited to *E. faecium*, and the evaluation was performed via VCM treatment for bacterial species such as penicillin-susceptible *E. faecium* that could be treated with penicillin antimicrobial agents [7]. To date, the effects of the disease status, concomitant agents, and pharmacokinetic/pharmacodynamic parameters, such as plasma trough concentration and AUC/MIC, on the clinical outcomes of VCM administration for the treatment of VCM-susceptible *E. faecium* have not yet been investigated.

This study aimed to investigate the clinical outcomes and evaluate the risk factors for mortality in adult patients treated for vancomycin-susceptible *E. faecium* bacteremia.

## 2. Materials and Methods

### 2.1. Study Population

This is a retrospective, record-based study. The data were collected from the inpatients of Juntendo University Hospital between January 2009 and December 2020. This study was conducted in accordance with the Helsinki Declaration, and approved by the Re-search Ethics Committee, Faculty of Medicine, Juntendo University (approval no. 20-270 and approval date of 18 December 2020).The inclusion criteria were as follows: the administration of VCM treatment in a catheter tip culture positive from *E. faecium* as a catheter-associated bloodstream infection; the administration of VCM treatment in at least two separate blood cultures positive from *E. faecium* as a bacteremia; the availability of plasma VCM concentration data; and age > 18 years. Only the first episode of *E. faecium* bacteremia was considered. The polymicrobial bacteremia patients with Gram-negative bacteria detected in a blood culture at the same time as *E*. *faecium* were only included in the analysis if they had received antimicrobial agents that were active in vitro against other coinfection pathogens.

The exclusion criteria were as follows: the administration of renal replacement therapy (hemodialysis or continuous hemodiafiltration); a lack of plasma VCM concentration data; the administration of antimicrobial agents (teicoplanin, linezolid, or daptomycin) other than VCM for the treatment of *E. faecium* bacteremia; the isolation of blood cultures with the simultaneous detection of Gram-positive bacterial species other than *E. faecium*; the isolation of blood cultures with Penicillin-susceptible *E. faecium*; the isolation of blood cultures with VCM-resistant *E. faecium* (VRE), when *E. faecium* was thought to be a contaminating bacterium; and a lack of MIC data.

The following patient data were extracted from medical records using predesigned forms: demographic characteristics (body weight, height, age, sex, underlying diseases, and admission to the intensive care unit); diagnosis of infectious diseases and source of infection by physicians; clinical laboratory data (levels of alanine aminotransferase, alkaline phosphatase, serum creatinine (SCr), and total bilirubin, as well as platelet and white blood cell counts); 30-day in-hospital mortality; dose of VCM administered; interval and initial plasma trough concentration of VCM at a steady state; concomitant agents; and vancomycin MIC of *E. faecium*. The patients were defined as having been diagnosed with septic shock if the doctor’s medical records included a description of septic shock. The onset of bacteremia was defined as the collection date of the first blood culture positive for *E. faecium*.

Regarding the drugs used before the onset of *E. faecium* bacteremia, the following cases were defined as “use”: the receipt of steroids and antineoplastic agents within 30 days before the positive blood culture or the receipt of any systemic antibiotics for >48 h in the month preceding the positive blood culture. In addition, in the history of antibiotic use defined above, if cephalosporins or carbapenems had been administered they were defined as “inactive antimicrobial agents against *E. faecium*”.

### 2.2. Calculation of AUC

The AUC of VCM from 0 to 24 h (AUC_24_) was calculated using the Bayesian approach, performed with a therapeutic drug monitoring software (Vancomycin MEEK TDM analysis software version 3.0; Meiji Seika Pharma Co., Ltd., Tokyo, Japan). The AUC_24_/MIC ratio was calculated using the AUC_24_ obtained for each patient and the MIC derived from the microbiological test results.

### 2.3. Measurement of MIC of E. faecium

The BACTEC FX system (Becton and Dickinson, Tokyo, Japan) was used to process the blood cultures. The *E. faecium* was isolated according to standard microbiological procedures. The isolates were identified using Microscan Walk Away (Beckman Coulter, Tokyo, Japan) and a matrix-assisted laser desorption/ionization time-of-flight mass spectrometry (MALDI-TOF MS; Bruker Daltonics, Kanagawa, Japan). The minimum inhibitory concentrations of vancomycin were determined with a microdilution method, using Microscan Walk Away (Beckman Coulter, Tokyo, Japan), and interpreted according to the breakpoints proposed by the Clinical and Laboratory Standards Institute guidelines.

### 2.4. Primary Endpoint

The primary outcome measure was the 30-day in-hospital mortality [7]. Moreover, the presence or absence of factors in the vancomycin-induced relation to renal dysfunction was also examined. Vancomycin-induced renal dysfunction was defined as a SCr rise of 0.3 mg/dL within 48 h or a 1.5-fold rise of SCr within 7 days, evaluated using the International Clinical Practice Guidelines (KDIGO) [8].

### 2.5. Statistical Analyses

The univariates were analyzed using the Mann–Whitney U test for continuous variables and the Fisher’s exact test for categorical variables. The multivariate analysis was performed on two items: the factors that influenced the 30-day in-hospital mortality and the concomitant agents associated with the onset of renal dysfunction during VCM administration. First, the correlation between the two variables was analyzed using Spearman’s ranking method. Second, a logistic regression analysis was performed using two factors with weak correlations. The alpha was set to 0.05 to evaluate statistical significance. The statistical analyses were compared using JMP Pro version 15.0 (SAS Institute Inc., Cary, NC, USA).

### 2.6. Consent to Participate

The patients’ consent was obtained by implementing an opt-out methodology.

## 3. Results

Forty-one patients diagnosed with *E. faecium* bacteremia during the study period were enrolled in this study (Table 1). The median age was 69 years, and the creatinine clearance calculated using the Cockcroft–Gault formula was 59.4 mL/min. Overall, 51.2% (21/41) of the patients had a history of admission to the intensive care unit, and 63.4% (26/41) of the patients had a malignant tumor as the underlying disease. Biliary tract infection was the most common disease (36.6%) among the patients. The most frequent MIC of vancomycin was ≤0.5 μg/mL (20 strains), followed by 1 μg/mL (16 strains). Nine of the forty-one patients (22.0%) died within 30 days of diagnosis with *E. faecium* bacteremia and were classified as part of the “30-day in-hospital mortality group”. Meanwhile, 32 patients were classified as part of the “survival group”.

The univariate analysis showed significant differences in the concomitant use of vasopressors, use of inactive antimicrobial agents against *E. faecium*, VCM plasma trough concentration, and renal dysfunction between the 30-day in-hospital mortality and survival groups during VCM administration (Table 2). However, the AUC_24_/MIC values in the two groups were not significantly different. Then, we performed a multivariate analysis on two factors of weakly correlated variables using Spearman’s ranking method, namely, the renal dysfunction during VCM administration and the concomitant use of vasopressors (*p* = 0.360). We identified the concomitant use of vasopressors as an independent risk factor for the 30-day in-hospital mortality (odds ratio, 7.81; 95% confidence interval [CI], 1.16–52.9; *p* = 0.035). This is shown in Table 3.

The group with renal dysfunction had a significantly higher concomitant use of vasopressor and piperacillin–tazobactam (PIPC/TAZ) (Table 4). We performed a multivariate analysis on two factors of weakly correlated variables using Spearman’s ranking method, namely, the vasopressor and the PIPC/TAZ (*p* = 0.185). We found that the combination of PIPC/TAZ with VCM may be an independent risk factor for renal dysfunction (odds ratio, 15.5; 95% confidence interval [CI], 2.04–117.6; *p* = 0.008) (Table 5).

## 4. Discussion

Previous studies report that the mortality in patients with *E. faecium* bacteremia was 25.0–34.6%, which is similar to the 30-day in-hospital mortality rate (22.0%) observed in this study [9,10]. In a previous retrospective study, the plasma VCM concentrations in enterococcal bacteremia patients were lower than those recommended for *S. aureus* bacteremia [6], and the mortality groups had lower plasma concentration levels than the survival groups [7]. They reported that an AUC_24_/MIC ratio of VCM > 389 might reduce mortality caused by enterococcal bacteremia [7]. In this study, the plasma trough concentrations and the AUC_24_/MIC ratio of VCM in the mortality group were higher than those in the surviving group, suggesting that the known plasma trough concentrations and AUC_24_/MIC ratio may not be associated with mortality in patients with *E. faecium* bacteremia. The time of evaluation of the AUC in the previous study was an average of 24 h during the initial 72 h [7]. However, we calculated AUC using the blood concentration of vancomycin at a single point, when the blood concentration reached a steady state. The difference in the time to calculate AUC may be a factor in differing results.

Another possibility is that VCM is bactericidal against most Gram-positive bacteria but bacteriostatic against enterococci [11]. Bacteriostatic antibacterial agents require phagocytic cells to eliminate bacteria and may be less effective in patients without a normal immune response [12]. Therefore, the known AUC_24_/MIC ratio ≥ 400 may not be a therapeutic target for enterococci. In our univariate analyses, the plasma VCM concentration, the renal dysfunction during VCM administration, the use of other antimicrobial agents inactive against *E. faecium* (cephalosporins or carbapenems), and the concomitant use of vasopressors were identified as factors affecting mortality. In our multivariate analyses, the concomitant use of vasopressors emerged as an independent factor affecting mortality. Malignancies, or immunosuppression impairing innate and adaptive immunity, appear to be the most significant risk factors for sepsis and septic shock [13]. The International Guidelines for the Management of Septic Shock 2016 recommend the use of vasopressors for septic shock when such patients do not respond to the initial fluid administration [14]. A study predicting the risk of VCM-induced acute kidney injury (AKI) reported that 32 (36.8%) out of 87 patients who received vasopressors developed AKI and were being treated [15]. AKI is linked to a higher risk of 30-day mortality in patients with enterococcal bacteremia [16].

According to the medical information records, eleven out of the forty-one target patients were diagnosed with septic shock, and eight were concomitantly treated with vasopressors. In the group of patients who died within 30 days, four out of the five patients who received a vasopressor agent developed renal dysfunction. Previous studies investigating the predictors of *S. aureus*-related mortality within 30 days have similarly reported that septic shock increases mortality, which is consistent with our results [17,18]. However, the concomitant use of vasopressors as a risk factor for *E. faecium* bacteremia is a novel finding of this study. One study investigated mortality risk factors in patients with hematologic malignancies who had bloodstream infections, and vasopressors were reported to be a significant factor (odds ratio, 6.06; 95% confidence interval [CI], 3.01–12.2; *p* < 0.001) [19]. Although the study’s population is different, it should be considered that the concomitant use of vasopressors in the setting of bacteremia carries a mortality risk. However, it should be kept in mind that our retrospective study with a small sample size may have revealed unjustified correlations and that the mortality rates between the groups were compared using a crude comparison method that was not adjusted for confounding variables.

A higher prevalence of sepsis has been reported in oncological patients vs. non-oncological patients [20]. Immunosuppression due to underlying malignancy or its treatment can increase the risk of severe infections [21]. Sixty-five percent of the cases included in this study had a background of immunosuppressive conditions such as solid tumors, hematological tumors, and liver transplantation, and all the cases in the 30-day mortality group had a background of malignant tumors. Therefore, when an immunocompromised host develops *E. faecium* bacteremia with septic shock, especially when a vasopressor is used in a patient with unstable hemodynamics, it may be difficult to treat it exclusively with clinical therapeutic exposure.

Furthermore, the combined use of PIPC/TAZ and VCM was identified as a risk factor for renal dysfunction. It has been previously reported that the concomitant use of PIPC/TAZ and VCM increases the risk of renal disorder compared to VCM monotherapy; our study results are consistent with the other literature [22,23]. However, other studies’ heterogeneity in the inclusion criteria and evaluation of outcomes using crude comparison methods which were not adjusted for confounding variables should be cautiously interpreted, as evidenced by the variability in our study.

This study has several limitations that are inherent to its retrospective design. The retrospective nature of our study should be considered for potentially raising issues regarding uncontrolled confounding variables such as potential selection bias, diagnostic bias, and misclassification bias. The composition of the patients included in our study may have limited the generalizability of the results. The number of cases was very limited due to the single university hospital chosen for the study and the target of vancomycin-susceptible *E. faecium* bacteremia cases.

Also, the AUC/MIC range of the target cases was wide. There are several possible reasons for this. First, the Bayesian method, which uses one-point trough concentrations, was used as the calculation method for the AUC24. There was an in-hospital protocol based on a 15 mg/kg/dose for 12 h, but, in some cases, vancomycin was administered at a dose based on the physician’s experience. This may have resulted in higher trough concentrations and larger AUC estimates. Second, the VCM clearance in the software used to calculate the AUC was calculated using the Cockcroft–Gault formula, and the population pharmacokinetic parameters built into this software fix a CL = 3.83 L/h when the Ccr ≥ 85 mL/min. Therefore, the AUC may be more inaccurate for these cases.

In addition, in the 30-day in-hospital mortality group, the high AUC/MIC may indicate that the systemic exposure increased due to a decrease in the vancomycin clearance course of the decreased renal function. It could not be measured accurately due to various factors such as the concomitant agents, the method of calculating the AUC, the decreased perfusion to the body’s organs associated with sepsis, or septic shock. For these reasons, we cannot exclude the possibility that residual bias affected our findings.

In conclusion, when an immunocompromised host develops vancomycin-susceptible *E. faecium* bacteremia with septic shock, especially when a vasopressor is used in a patient with unstable hemodynamics, they may be difficult to treat with sufficient antimicrobial exposure alone. VCM was the most effective drug for vancomycin-susceptible *E. faecium*. Therefore, it is necessary to consider concomitant drugs and provide suitable treatment individually, for each patient, while monitoring blood concentrations to avoid the onset of renal dysfunction.

## Figures and Tables

**Table 1 antibiotics-12-01577-t001:** Patients’ characteristics.

Characteristics	Value
**Sex, *n***	
Male	24
Female	17
Age, year, median (IQR)	69 (58–79)
Body weight, kg, median (IQR)	52.9 (46.0–59.9)
**ICU**	
Number of ICU patients admitted during hospitalization, *n* (%)	21 (51.2)
ICU admission period (day), median (IQR)	5.0 (3.3–18.0)
**Laboratory results**	
Serum creatinine level, mg/dL, median (IQR)	0.73 (0.60–1.00)
Creatinine clearance, mL/min, median (IQR) ^a^	59.4 (40.7–100.9)
Alanine aminotransferase level, IU/L, median (IQR)	42 (16–97)
Total bilirubin level, mg/dL, median (IQR)	1.34 (0.72–3.25)
White blood cell count, ×10^9^/L, median (IQR)	9.6 (5.6–12.2)
Platelet count, ×10^9^/L, median (IQR)	131 (82–176)
Alkaline phosphatase level, IU/L, median (IQR) ^b^	412.5 (237.5–877.3)
**Underlying disease, *n* (%)**	
Solid malignancy	22 (53.7)
Hematologic malignancy	4 (9.8)
Central nervous system disease	4 (9.8)
Cardiovascular disease	4 (9.8)
Liver cirrhosis	2 (4.9)
Hepatic transplantation	1 (2.4)
Respiratory disease	1 (2.4)
Other diseases ^c^	3 (7.3)
**Infection**	
Hepatobiliary system infection	15 (36.6)
Intra-abdominal infection	7 (17.1)
Urinary tract infection	5 (12.2)
Catheter-related bloodstream infection	5 (12.2)
Febrile neutropenia	4 (9.8)
Infective endocarditis	2 (4.9)
Other infections ^d^	3 (7.3)
**VCM-MIC of *Enterococcus faecium*, μg/mL, *n* (%)**	
≤0.5	20 (48.8)
1	16 (39.0)
2	5 (12.2)

^a^ Calculated using the Cockcroft–Gault formula of t-Gault. ^b^
*n* = 32. ^c^ Other diseases are granulomatosis with polyangiitis and Münchhausen syndrome. ^d^ Other infections are pneumonia, iliopsoas abscess, and wound infection. VCM, vancomycin; IQR, interquartile range; ICU, intensive care unit; and MIC, minimum inhibitory concentration.

**Table 2 antibiotics-12-01577-t002:** Comparison between the 30-day in-hospital mortality and survival groups using univariate analysis.

Variable	Survival (*n* = 32)	30-Day in-Hospital Mortality (*n* = 9)	*p*
**Disease status, *n* (%)**			
Solid tumor	18 (56.3)	4 (44.4)	0.71 ^a^
Diabetes mellitus	5 (15.6)	3 (33.3)	0.34 ^a^
ICU admission	15 (46.9)	6 (66.7)	0.45 ^a^
Use of central intravenous catheter	12 (37.5)	6 (66.7)	0.15 ^a^
Hepatobiliary system infection	12 (37.5)	3 (33.3)	1.00 ^a^
Intra-abdominal infection	7 (21.9)	0 (0.0)	0.31 ^a^
Urinary tract infection	3 (9.4)	2 (22.2)	0.30 ^a^
Catheter-related bloodstream infection	2 (6.25)	3 (33.3)	0.06 ^a^
Febrile neutropenia	4 (21.9)	0 (0.0)	0.56 ^a^
Infective endocarditis	2 (6.3)	0 (0.0)	1.00 ^a^
**Use of concomitant agents, *n* (%)**			
Thrombomodulin	3 (9.4)	0 (0.0)	1.00 ^a^
Vasopressor	3 (9.4)	5 (55.6)	0.007 ^a^
Globulin	2 (6.3)	2 (22.2)	0.20 ^a^
Total parenteral nutrition	14 (43.8)	3 (33.3)	0.71 ^a^
**Medication history, *n* (%)**			
Steroids ^c^	9 (28.1)	4 (44.4)	0.43 ^a^
Anticancer agents ^c^	7 (21.9)	2 (22.2)	1.00 ^a^
Antimicrobial agents that are inactive against *E. faecium*, including cephalosporins and carbapenems ^d^	13 (40.6)	8 (88.9)	0.02 ^a^
Antimicrobial agents ^d^	15 (46.9)	8 (88.9)	0.05 ^a^
**VCM administration, median (IQR)**			
Dose, mg/day	1250 (1000–2000)	1000 (750–1000)	0.16 ^b^
**Pharmacokinetic/pharmacodynamic parameters, median (IQR)**			
Plasma trough VCM concentration, mg/L	9.1 (6.6–11.9)	14.9 (11.0–19.5)	0.01 ^b^
AUC_24_/MIC	658.5 (427.1–990.5)	852 (481.8–1558.0)	0.28 ^b^
MIC	1.0 (0.5–1.0)	0.5 (0.5–1.0)	0.96 ^b^
Trough level ≤ 15 µg/mL	27(84.4)	5 (55.6)	0.08 ^a^
Trough level > 15 µg/mL	5 (15.6)	4(44.4)	
AUC_24_/MIC < 389 n(%) (Range)	6 (18.7) (165.7–387.2)	2 (22.2) (237.2–307.2)	1.00 ^a^
AUC_24_/MIC ≥ 389 n(%) (Range)	26 (81.3) (402.1–2070.4)	7(77.8) (481.8–2005.2)	
Renal dysfunction during VCM administration	6 (18.8)	6 (66.7)	0.01 ^a^

^a^ Fisher’s exact test. ^b^ Mann–Whitney U test. ^c^ Number of uses within 30 days before a positive blood culture test. ^d^ Continuous administration for ≥48 h a month before a positive blood culture. AUC, area under the curve; VCM, vancomycin; IQR, interquartile range; and ICU, intensive care unit.

**Table 3 antibiotics-12-01577-t003:** Univariate and multivariate analyses of the factors associated with the 30-day in-hospital mortality.

Factors	Univariate Analysis			Multivariate Analyses			Spearman’s Rank Correlation Coefficient
	Odds Ratio	95% CI	*p*	Odds Ratio	95% CI	*p*	
Renal dysfunction during VCM administration	8.6	1.67–44.9	0.011	5.65	0.94–34.1	0.059	0.360
Concomitant use of vasopressor	12.1	2.05–71.1	0.007	7.81	1.16–52.9	0.035	
Antimicrobial agents that are inactive against *E. faecium*, including cephalosporins and carbapenems	11.7	1.30–105.0	0.021	–	–	–	

CI, confidence interval; VCM, vancomycin.

**Table 4 antibiotics-12-01577-t004:** Comparison between the groups with and without renal dysfunction during VCM administration.

Variable	Without Renal Dysfunction (*n* = 29)	With Renal Dysfunction (*n* = 12)	*p*
**Disease status, *n* (%)**			
Solid tumor	16 (55.2)	6 (50.0)	1.00 ^a^
Digestive system tumor	14 (48.3)	6 (50.0)	1.00 ^a^
Hepatobiliary tumor	8 (27.6)	5 (41.7)	0.47 ^a^
Diabetes mellitus	5 (17.2)	3 (25.0)	0.67 ^a^
ICU admission	6 (20.7)	3 (25.0)	1.00 ^a^
Use of central intravenous catheter	11 (37.9)	7 (58.3)	0.31 ^a^
Hepatobiliary system infection	10 (34.5)	5 (41.7)	0.73 ^a^
Intra-abdominal infection	5 (17.2)	2 (16.7)	1.00 ^a^
Urinary tract infection	4 (13.8)	1 (8.33)	1.00 ^a^
Catheter-related bloodstream infection	2 (6.9)	3 (25.0)	0.14 ^a^
Febrile neutropenia	3 (10.3)	1 (8.33)	1.00 ^a^
Infective endocarditis	2 (6.9)	0 (0.0)	1.00 ^a^
**Use of concomitant agents, *n* (%)**			
Thrombomodulin	3 (10.3)	0 (0.0)	0.54 ^a^
Vasopressor	3 (10.3)	5 (41.7)	0.03 ^a^
Globulin	2 (6.9)	2 (16.7)	0.56 ^a^
Total parenteral nutrition	13 (44.8)	4 (33.3)	0.73 ^a^
Diuretic	7 (24.1)	7 (58.3)	0.07 ^a^
PIPC/TAZ	2 (6.9)	7 (58.3)	0.001 ^a^
**Medication history, *n* (%)**			
Steroids ^c^	11 (32.4)	1 (20.0)	1.00 ^a^
Anticancer agents ^c^	9 (26.5)	0 (0.0)	0.31 ^a^
Antimicrobial agents inactive against *E. faecium*, including cephalosporins and carbapenems ^d^	17 (50.0)	2 (40.0)	1.00 ^a^
Antimicrobial agents ^d^	19 (55.9)	2 (40.0)	0.65 ^a^
**VCM administration, median (IQR)**			
Duration, day	15 (13–24)	13.5 (7–16.5)	0.22 ^b^
Dose, mg/day	1000 (1000–2000)	1000 (937.5–2000)	0.81 ^b^
**Pharmacokinetic/pharmacodynamic parameters, median (IQR)**			
Plasma trough VCM concentration, mg/L	9.0 (6.8–12.8)	13.3 (9.2–18.1)	0.10 ^b^
AUC	454.7 (411.2–617.1)	529.6 (452.8–763.4)	0.34 ^b^

^a^ Fisher’s exact test. ^b^ Mann–Whitney U test. ^c^ Number of uses within 30 days before a positive blood culture test. ^d^ Continuous administration for ≥48 h a month before a positive blood culture. AUC, area under the curve; VCM, vancomycin; IQR, interquartile range; and PIPC/TAZ, piperacillin–tazobactam.

**Table 5 antibiotics-12-01577-t005:** Univariate and multivariate analyses of factors affecting renal dysfunction in the patients administered with VCM.

Factors	Univariate Analyses			Multivariate Analyses			Spearman’s Rank Correlation Coefficient
	Odds Ratio	95% CI	*p*	Odds Ratio	95% CI	*p*	
Concomitant use of vasopressor	6.2	1.18–32.5	0.035	6.2	0.84–45.2	0.074	0.185
Concomitant use of PIPC/TAZ	18.9	3.01–118.8	0.001	15.5	2.04–117.6	0.008	
Concomitant use of diuretic	4.4	1.05–18.4	0.068	–	–	–	

CI, confidence interval; PIPC/TAZ, piperacillin–tazobactam; and VCM, vancomycin.

## Data Availability

The datasets generated and analyzed during the current study are available from the corresponding author upon reasonable request.

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
