# Peer review of "Assessing Clinical Outcomes of Vancomycin Treatment in Adult Patients with Vancomycin-Susceptible Enterococcus faecium Bacteremia"

_antibiotics, 2023, doi:10.3390/antibiotics12111577_

Round 1

Reviewer 1 Report

Comments and Suggestions for Authors

General comment: This work is a retrospective clinical study that aimed to investigate the clinical outcomes and risk factors for mortality in adult patients with vancomycin-susceptible E. faecium bacteraemia.

This work as several limitations in the study design as well as several problems with English grammar, that makes very difficult to understand some sentences over the manuscript.

Study design: the exclusion criteria includes administration of other antimicrobial agents rather that vancomycin, but then the authors describe that included patients that were submitted to several other antimicrobial agents; what do you mean by 30-day all-cause mortality (line 83); why did you use a non-parametric test to evaluate continuous variables in the statistical analysis?

Grammar problems in the following sentences: Lines 39-42; 52-53;124-126;

Also, the bacterial taxonomic rules are not followed in several points of the manuscript (missing italics on lines 46, 66, 76, 77, 98, and correct bacterial species on line 76).

Missing references in lines 34, 42, 48, 147, 165.

Specific comments:

Introduction section

Please change flora and microflora to microbiota (lines 34 and 37)

Please rewrite the goal of the study on lines 52-53, it is confused.

Methods section

Why did you used the homepage of the hospital to perform the informed consent of the study? This is not acceptable! First, how do you guarantee that all participants use the internet or have access to it? Also, it is confidential information that you are working.  (lines 61 and 113-117). You may use a program to get the data and do an anonymization of the data, in order to guarantee that no personal information or identification of the patient can occur.

In the study design there are several contradictory information regarding the use or not of other antimicrobials. This is a critical point.    (lines 69, 74-77, 89; 130)

In line 98 you must describe what antimicrobial are you testing for MIC.

Please include the software used to perform the statistical analysis and explain the evaluation performed and presented in table 3 and 4 using odds ratio.

Results section

Table 2, please revise the information regarding the AUC/MIC, that is contradictory. If the range was 427.1-990.5 and 481.8-1558.0, how do you identify 6 cases and 2 cases of AUC/MIC inferior to 389???

Also, in table 2, you included information regarding several antimicrobial agents that is not in agreement with the methods.

Table 3 is not clear regarding the ODDS ratio analysis and what was done.

Discussion section

Where do you describe the septic shock as a diagnose? I only found that information on discussion.

What in “pressor”? (lines 173 and 185)

Revise sentence 163-164 and 195.

Comments on the Quality of English Language

This work as several problems with English grammar, that makes very difficult to understand some sentences over the manuscript ( Lines 39-42; 52-53;124-126)

Author Response

Dear reviewer

We have revised the manuscript according to your comments.

親愛なるレビュアー様

いただいたコメントをもとに原稿を修正しました。

添付ファイルをご覧ください。

Reviewer 2 Report

Comments and Suggestions for Authors

Very interesting article with some early data for clinical use.  The retrospective nature of the study design has some concerns and limitations that need to be addressed and should be reflected in the conclusions that are drawn.

Abstract

Line 18  what is the meaning of inactive antimicrobials --are you referring to antibiotic resistance?  Clarify statement

Line 22  there is significant variability in the range of 1.15 to 52.9.  As a researcher when I see this type of variability, my first reaction is multiple factors or confounders are involved and it is not solely explained by the risk factor discussed in the paper.  The 7.81 odds risk maybe overinflated due to the variability with the retrospective design and the small number of patients.  I would be very careful about drawing too many conclusions solely from this odds risk. 

Line 27 & 28  Throughout the paper, the statement using unstable is not clear.  "especially in situations where the hemodynamics of using a pressor agent is unstable,"  Are the hemodynamics unstable for the patients versus is the pressor agent unstable?  Why would a pressor agent be unstable versus using a pressor agent in a patient with unstable hemodynamics, which fits the picture of critical care and sepsis. 

Line 25 to 29  Conclusion.  Be very careful with this conclusion.  Indicate potential risk factors that contribute mortality in these patients BUT just because you saw no association with AUC/MIC does not mean we do not try or need to make sure we achieve adequate AUC/MIC and therapeutic vanco levels.  

Introduction 

Line 42 to 43 Incomplete sentence  However, this target ratio is considered for the suspected or definitive diagnosis of MRSA infections.  Do you mean This target ratio is a therapeutic goal with suspected or definitive diagnosis of MRSA?  Delete the However

Line 44 delete that

Line 45 delete that and replace with the

Line 52 change from we aimed  replace In this study, the objective was to investigate....

Line 72  add was    when E. faecium was thought to be 

Line 75 change that to with

Line 82  add abbreviation here so it says  serum creatinine (SCr)

Then change all other abbreviations to SCr in the text

Line 86 delete that was

Line 90  48 h/month, please spell out and then explain what you mean by this   Is it 48 hours within the month???

Line 94 delete Thereafter

Line 98  need a better explanation here   Was the MIC for all samples done by the same lab with the same methods over the study time frame.  Did they purchase the MIC plates from the same company? Did the lab change any aspect during the study time?

Line 101  vancomycin should be lower case

Line 102 should that be defined as versus defained  should it read was defined as SCr rise of 0.3 mg/dL   (correct SCr twice inthis paragraph)

Line 112  You do not set a P value.  You set alpha and evaluate your P value in relation to alpha. It should read.  Alpha was set at 0.05 to evaluate statistical significance.   

Line 121  Why Cockcroft-Gault versus other formulas since values can be unreliable in critical care patients especially sepsis?

Line 123-124 What are you trying to say?  The VCM MIC value was the ?? Do you mean  The most frequency VCM MIC was ...... (20 strains), followed by

Line 145  Who reports mortality --is it previous studies, literature review.  Start with the studies.  Previous studies report.....

Line 150  remove also

Line 151 remove However

Line 156 extra period needs removed after reference 11

Line 161  So later in article, you refer to pcn-resistant (Line 193)--this is the first reference to this --it is not in your abstract, introduction, or inclusion criteria.  In Line 161, you mention cephalosporins for the first time but the rest of the article refers to Pip Tazo, which is a penicillin derivative.  Very confusing, this needs to be addressed in the entire article or some explanation of why you are flipping between pcn, pip tazo, and cephalosporins. 

Line 164  I would suggest deleting due to a malignant tumor or the like.  Definitely remove "or the like"  I think is sufficient to indicate immunosuppression.   Sepsis tends to become severe and cases of shock needs to be referenced.  What supports this statement. 

Line 167 Remove Moreover

Line 169 - 170  Is a sentence missing--seems incomplete 

Line 172 remove coma after shock

Line 175 remove that

Line 180 needs a space after reference 19. before sixty-five 

Line 184 and 185 Unstable again as an issue?  Also consider changing last part of sentence to  it may be difficult to treat exclusively by clinical therapeutic exposure ?  versus sufficient 

Line 188 remove Although

Line 189 sentence needs clarification .  VCM has been previously reported, our study results are consistent with this literature (something to this effect needs added)

Line 191  There are more limitations to the study beyond what has been presented.  The retrospective nature creates problems with confounding variables not controlled.  Other confounding issues explaining the variability i odds risk.  Potential selection bias and diagnosis bias.  Misclassification bias.

Line 193  PCN issue being first mentioned, see previous comment  Should this be included in the inclusion criteria

Line 196 remove However  should the word in this line be changed from mentioned to measured

Line 200  conclusion indicated pcn-resistance --where is this in the methods, introduction.  why is it coming into the conclusion now?  Justify

Line 201  unstable issue again

Line 203 However remove.  conclusion of VCM was the most effective drug for pcn-resistant E. faecium     Did not justify this in introduction or by the study.  This is a broad conclusion not supported by the study--in conclusion address the findings supporting primary outcome.  Stay to your study. 

Line 205  should it be concentrations versus concentration

Reference formats needed corrected.

Why is the odds risk and variability not mentioned in the discussion section.  Develop this more with more discussion of table results that are important to discuss. 

Comments on the Quality of English Language

Minor issues, will address in author section.

Author Response

Dear reviewer

We have revised the manuscript according to your comments.

Reviewer 3 Report

Comments and Suggestions for Authors

The manuscript describes a retrospective study to assess clinical outcomes of vancomycin therapy. Given the objectives of the study and drawbacks associated with retrospective studies, a better design should be a prospective study. What was the rationale for that?

Few points which need to be addressed are:

In abstract and study methodology, please mention the study design as retrospective, record-based study.

Regarding informed consent, the authors mention about displaying the information regarding study at website and giving opportunity to patients to refuse to participate, thereby meaning “implied consent”. But whether any effort was made to confirm about what proportion of patients actually received the information. Were there any refusals to participate?

Please re-phrase to make it clear “Polymicrobial bacteremia patients that gram-negative bacteria detected from blood culture at the same time as E.faeiucm were only included in the analysis if they had received antimicrobial agents that were active in vitro against other coinfection pathogens”.

Re-phrase “Vancomycin-induced renal dysfunction was defained that SCR rise of 0.3 mg/dL or more within 48 hours, or a 1.5-fold rise of SCR within 7 days, by evaluated using the International Clinical Practice Guidelines”.

Please provide any data, if available, with respect to incidence of E.fecium isolates from bloodstream and other infections at the study site in introduction. Also, during the study period, what was the incidence of isolation of E. faecium?

The inclusion criteria are mentioned as “started treatment as a catheter-associated bloodstream infection at catheter tip culture positive, or at least two separate blood cultures positive from E. faecium”. It seems the study population comprised of both catheter-associated i.e. healthcare associated infections as well as community acquired infections. If that is the case, please clarify and give rationale for the same as the choice of antibiotics in the two scenarios usually do not coincide.   

Out of 64 eligible patients, 41 were included. What were the reasons for exclusion of remaining 23 patients?

Tables 3 and 5 represent the results of univariate and multivariate analyses for only 2 variables while in methods, other variables are also listed. Please add the result for all the variables analysed even though they may be non-significant.

Is there any typographical error?  Table 2, under mortality group, VCM dose range (750-100?)

What was the protocol of VCM administration regimen? The median dose of VCM in the 2 groups (survival vs mortality) is quite different.  

Discussion, 1st paragraph, the authors discuss about contradictory findings compared to an earlier study. Please discuss this a bit more elaboratively and reasons for the same.

Comments on the Quality of English Language

Moderate editing required.

Author Response

(The authors gave the same response as above.)

Round 2

Reviewer 2 Report

Comments and Suggestions for Authors

Thank you for thoughtful consideration of suggested edits.